# Sirtuin 1 Expression in Canine Mammary Tumors: A Pilot Study

**DOI:** 10.3390/ani13162609

**Published:** 2023-08-13

**Authors:** Mariafrancesca Sgadari, Nunzio Antonio Cacciola, Karen Power, Manuela Martano, Brunella Restucci

**Affiliations:** Department of Veterinary Medicine and Animal Productions, University of Naples “Federico II”, Via F. Delpino 1, 80137 Naples, Italy; nunzioantonio.cacciola@unina.it (N.A.C.); karen.power@unina.it (K.P.); manuela.martano@unina.it (M.M.)

**Keywords:** sirtuin family, canine mammary tumors, veterinary oncology, culture cells

## Abstract

**Simple Summary:**

Sirtuin family proteins (SIRTs) are intra-cellular enzymes that are found in mammals. They have a regulatory function in genomic transcription and are involved in a variety of physiologic mechanisms, such as aging and genome stability. Of these enzymes, SIRT1, the most investigated, may also be involved in cancerogenesis, although its possible dual role as an onco-suppressor and/or tumor promoter under a variety of biological conditions in humans and animals is still controversial. In normal or neoplastic canine mammary tissues, SIRT1 has not yet been investigated. We examined the cellular and subcellular distribution of SIRT1 in non-neoplastic (*n =* 5) and neoplastic (*n =* 45) mammary tissue samples of dogs using immunohistochemistry. In addition, we examined the expression and subcellular localization of SIRT1 in cultured canine mammary cancer cells by Western blot and immunofluorescence analyses. We found abundant and mainly nuclear expression in normal glandular cells; expression progressively weakened in mammary cancer cells in correlation with histologic features of increasing malignancy. In addition to weakening expression, a shift in the subcellular SIRT1 expression from predominantly nuclear to largely cytoplasmatic was also observable. In cultured canine cancer cells, subcellular localization was cytoplasmatic throughout. Our results suggest that SIRT1 may play a tumor0protecting role in canine mammary tumors, as deductible from the shifts in subcellular localization.

**Abstract:**

Sirtuin 1 (SIRT1) is a protein involved in aging, cell protection, and energy metabolism in mammals. Recently, SIRT1 has been intensively studied in medical oncology, but the role of SIRT1 is still controversial, as it has been proposed as both an oncogene and a tumor suppressor. The aim of this study is to investigate the expression of SIRT1 by immunohistochemistry in canine mammary tissues, and by Western blot and immunofluorescence analysis in different canine mammary cell lines. Our results showed a decrease in SIRT1 expression from normal mammary gland tissue, and from benign and well-differentiated malignant tumors (G1) to less differentiated ones (G2–G3). Furthermore, a shift in the subcellular localization of SIRT1 from the nucleus to the cytoplasm was observed in less differentiated malignant tumors. However, further studies are needed to investigate the subcellular localization of SIRT1 in canine cancer cells and the role it may play in oncogenesis in animals.

## 1. Introduction

Silent Information Regulator 2 (SIR2) proteins (called Sirtuins—SIRTs) belong to a family of regulatory enzymes, encoded by the SIR2 gene, which are found in most organisms, including mammals [1]. Sirtuins play a central role in transcriptional control as gene promotors through histone modification. They are highly conserved nicotinamide adenine dinucleotide (NAD+)-dependent histone deacetylases, also known as class III HDACs; they are actively involved in heterochromatin formation, gene silencing, metabolism, and aging [2,3,4,5]. Originally, they were mainly considered to be regulatory proteins in cellular mechanisms of survival and longevity, modifiers of a variety of subcellular substrates [6], and players in genome stability via transcriptional silencing and chromatin remodeling [7]. The sirtuin family consists of seven mammalian proteins, each of which has its own cellular sub-localization and molecular pathway. Of these, Sirtuin 1 (SIRT1) is one of the most studied. It is found in both the nucleus and cytoplasm interacting with protein substrates in a variety of signal transduction pathways. More recently, SIRT1 has been reported to be a critical regulator for cell metabolism in cancer development and progression [8]. In fact, several studies assign a dual role to SIRT1 in cancerogenesis [9,10]. While some investigators have reported an onco-suppressive effect of SIRT1, others have demonstrated the opposite [8]. However, this contradictory role seems to depend on the cell type, the stage of tumor development, and the subcellular localization of SIRT1 [10]. In veterinary oncology, a single case of decreased expression of SIRT1 in tumor cells compared to peripheral blood cells was reported in a single case and appeared to be linked to p53 gene mutation [11]. To the best of our knowledge, the expression of SIRT1 in canine mammary tumors has not yet been investigated. We, therefore, aimed to study possible SIRT1 involvement in different types of canine spontaneous mammary tumors (CMTs) and in different cultured CMT cell lines, employing a model previously published [12,13]. Canine spontaneous mammary tumors are considered a useful animal model for studying the molecular mechanisms underlying malignant transformation [14,15] and may be useful for improving the prognosis and treatment of human breast cancer (HBC) [16]. Indeed, canine cancer cell lines have recently been successfully used as models to study the biology of HBC [17,18], to investigate molecular signaling pathways implicated, and to test the efficacy of anticancer drugs. Therefore, the aim of the present study was to investigate the expression of SIRT1 canine mammary cell lines by Western blot analysis (WB). We also aimed to use immunohistochemistry (IHC) to identify SIRT1 expression in normal canine mammary tissues and in different CMTs. In these samples, we aimed to investigate cellular SIRT1 immunoexpression in relation to the different histological malignancy grades. In addition to IHC, immunofluorescence was used to further determine SIRT1 subcellular localization.

## 2. Materials and Methods

### 2.1. Canine Mammary Tissue Collection for Histology

Forty-five tissues were harvested from spontaneous CMTs, which were surgically removed in accordance with a routine treatment protocol at the Veterinary Teaching Hospital of the Department of Veterinary Medicine and Animal Production of Naples Federico II University; sampling for histologic diagnosis and treatment followed Directives 2010/63/EU. In addition, 5 macroscopically and histologically non-neoplastic (“normal”) mammary gland (NMG) tissue samples were used as controls. These samples were the remaining non-neoplastic mammary glands of dogs in which the entire mammary chain had been removed.

All samples were divided into two aliquots and stored under appropriate conditions according to the analytic protocol to be performed. Histology and immunohistochemistry were carried out on specimens fixed in 10% neutral buffered formalin and embedded in paraffin. Histologic diagnosis was performed on slides stained with hematoxylin and eosin, according to the updated classification and criteria of the Davis–Thompson DVM Foundation [19]. Histological grading of the tumors was performed according to the criteria proposed by Peña [20]. Evaluation of tubule formation, nuclear pleomorphism, and number of mitoses was done and averaged per 10 high power fields (HPF). According to these criteria, tumors were classified into four groups: benign tumors (BT) (10/45), well-differentiated (G1) carcinomas (12/45), moderately differentiated (G2) carcinomas (12/45), and poorly differentiated (G3) carcinomas (11/45). 

### 2.2. Immunohistochemistry

Tissue sections were deparaffinized in xylene, dehydrated in graded alcohols, and washed in 0.01 M PBS pH 7.2–7.4. Endogenous peroxidase was blocked with 0.3% hydrogen peroxide in absolute methanol for 30 min. Immunohistochemistry was performed by the streptavidin–biotin–peroxidase complex method using a commercially available kit as described elsewhere [13] (streptavidin–biotin–peroxidase method LSAB kit; Dako, Glostrup, Denmark). For this study, the available primary antibody against SIRT1 (mouse anti-human monoclonal antibody, 1:100, Biorbyt, Cambridge, UK, orb306144) was used. The immunolabeling procedure also included negative control sections incubated with normal serum IgG (Dako) instead of the primary antibody. A sample of canine kidney was used as a positive control. Immunoreactivity was assessed by two pathologists (BR and MS) and graded according to the number of positive cells in 10 High Power Fields (40× objective and 10× ocular; grade 0: no positive cells; 1: 10%; 2: 10.1–30%; 3: 30.1–60%; 4: > 60.1%), and the intensity of staining was classified as weak (1), moderate (2), or strong (3). Then, a combined immunoreactivity score (IR score) was calculated for each sample by multiplying the values of these two categories, ranging from 0 to 12 according to previous studies [13,21,22].

### 2.3. Cell Lines and Cultures

Six canine mammary tumor cells were used for this study, namely, CMT-U309 (spindle cell carcinoma cell line), CMT-U27 (simple ductal carcinoma cell line), P114 (anaplastic carcinoma cell line), CMT-U229 (atypical benign mixed tumor cell line), CMT-U131 (infiltrating ductal carcinoma of scirrhous type), and CF33 (adenocarcinoma cell line). In addition, canine MDCK non-cancerous kidney epithelial cells were used as a control, which has been reported previously [23]. The cross-reactivity of the antibody used was confirmed with the following human cell lines: MCF 10A (healthy human breast epithelial cell line), MDA-MB-231 (human triple negative breast cancer cell line), MCF7 (human breast adenocarcinoma cell line), Hep G2 (human hepatocellular cancer cell line), and HeLa (human cervical cell line). The human cancer cell lines were obtained from the American Type Culture Collection (ATCC, Manassas, VA, USA). The cell cultures were started in the same medium recommended by ATCC for the best results. Accordingly, we were aware that the cell lines analyzed did not change their target expression. Canine mammary cell lines CMT-U131, CMT-U309, CMT-U27, and CMT-U229 were cultured in Roswell Park Memorial Institute medium (RPMI-1640), while MDCK, P114, and CF33 were cultured in Dulbecco’s Modified Eagle Medium (DMEM), supplemented with 10% (*v/v*) heat-inactivated and filtered fetal bovine serum (FBS), plasmocin (50 iU mL^−1^), and 2 mM L-glutamine (all reagents are from Sigma Aldrich, St. Louis, MO, USA). All cell lines were incubated in an atmosphere of 5% CO_2_ and 95% humidified air at 37 °C and routinely subcultured every other day. All cell lines were routinely tested for mycoplasma contamination.

### 2.4. Western Blot Analyses

For preparation of whole cell extracts, the above cell lines were washed twice in PBS and then lysed in a lysis buffer containing 20 mM Tris-HCl (pH 7.5), 150 mM NaCl, 1 mM Na_2_EDTA, 1% NP-40, 1% sodium deoxycholate), and a protease/phosphatase inhibitor cocktail (Cell Signaling Technologies, Cat # 5872). Lysates were sonicated and then clarified by centrifugation at 12,000× *g* for 10 min at 4 °C. Protein concentration was determined using the DC protein assay (Bio-Rad, Milan, Italy). Equal amounts of protein (40 μg) were loaded into 8% sodium dodecyl sulfate–polyacrylamide gels. The separated proteins were transferred to nitrocellulose membranes, which were then blocked for 1 h in 5% nonfat milk in Tween–Tris-buffered saline (TTBS) solution containing 25 mM Tris (pH 7.4), 150 mM NaCl, and 0.1% Tween 20. Membranes were probed overnight at 4 °C with the following primary antibodies: anti-SIRT1 (Biorbyt, orb69300, mouse monoclonal antibody, dilution 1: 1000) and anti-β-actin (Santa Cruz Biotechnology Inc., Dallas, MA, USA, #sc-47778, dilution 1:20,000). After incubation of the primary antibodies, the membranes were washed 3 times for 5 min, each time with fresh 1× TTBS. After the washing steps, the membrane was blotted with anti-rabbit or anti-mouse secondary antibody solution for 1 h at room temperature. After 3 wash steps in 1× TTBS, immunoreactive bands were visualized on the blot using an Enhanced Chemiluminescence Kit followed by exposure to X-ray film. Protein bands were quantified by densitometry using ImageJ software 1.53t (National Institutes of Health, Bethesda, MD, USA).

### 2.5. Immunofluorescence

In 24-well plates, 2 × 10^4^ cells per well of each cell line were seeded using a sterile 13 mm coverslip (Nunc™ Thermanox™, Cat. 174950). After 24 h of incubation in an atmosphere of 5% CO_2_ and 95% humidified air at 37 °C, cells were washed three times with cold PBS and fixed with 4% paraformaldehyde in PBS pH 7.4 for 15 min at room temperature. After washing three times with PBS for 5 min, the samples were stored in a blocking solution (containing 1% BSA, 22.52 mg/mL glycine in PBST (PBS + 0.1% Tween 20)) for 60 min at room temperature. The primary anti-SIRT1 antibody (Biorbyt, orb69300, mouse monoclonal antibody) diluted 1:100 in PBS + 1% BSA was applied overnight at 4 °C in a humidity chamber and incubated overnight. The next day, secondary anti-mouse antibody conjugated with Alexa Fluor™ Plus 555 (A32727, Thermo Fisher Scientific, Inc., Waltham, MA, USA, Goat anti-Mouse IgG (H+L)) and diluted 1:200 in 1% bovine serum albumin in PBS was applied for 2 h in the dark and humid chamber. After washing three times, the coverslips were fixed on the slide with the embedding medium (Dako ™). The cells were imaged under a laser scanning microscope (Leica ™ Microsystems, Germany). Mouse anti-SIRT1 monoclonal antibody was illuminated at 543 nm and measured using a 560 nm bandpass filter. Data acquisition and analysis were performed using the software LAS X (version 3.1.2.16221). To quantitatively evaluate the intensity of the fluorescence signal, the obtained images of the individual cells were analyzed using ImageJ software, available online at https://imagej.nih.gov/ij/download.html accessed on 14 June 2023), to calculate the Corrected Total Cell Fluorescence (CTCF) value [24].

### 2.6. Statistical Analysis

Results were expressed as mean ± standard error (SEM). Data were analyzed using GraphPad Prism v9.2.0 software (La Jolla, CA, USA). One-way ANOVA followed by Holm–Šídák’s multiple comparisons test to compare differences between mean values of SIRT1 IR. All data were tested for normality distribution using Kolmogorov–Smirnov test for all variables. Results with *p* < 0.05 were considered statistically significant.

## 3. Results

### 3.1. Immunohistochemistry

According to the IHC results, only ductal and lobular epithelial cells showed positive SIRT1 immunostaining, both in the NMGs and CMTs; that is, in five out of five NMGs (100%), a strong nuclear expression (mean immunoreactivity IR score = 9.8 ± 1.562, range 4–12) was found in epithelial ductal cells (Figure 1A). All benign tumors (10/10; 100%) showed strong immunostaining (mean IR score = 8.50 ± 1.40), mainly localized in the nucleus (Figure 1B). All well-differentiated (G1) carcinomas (12/12; 100%) displayed strong SIRT1 immunoreactivity (mean IR score = 6.16 ± 0.936, range 2–12) and nuclear immunostaining, even if a moderate cytoplasmic reactivity was found in some neoplastic cells (Figure 1C). Although all moderately differentiated (G2) carcinomas (12/12; 100%) showed a progressive decrease in IR score values and a loss of nuclear localization for SIRT1 (mean IR score = 5.50 ± 0.690, range 2–9), the protein was restricted to regions where the mammary glandular morphology was still preserved (Figure 1D). Furthermore, 9 out of 11 poorly differentiated (G3) carcinomas (81.81%; mean IR score = 3.00 ± 0.713, range 0–6) stained only weakly for SIRT1, though moderate immunostaining was present in the phenotypically less aggressive counterparts (Figure 1E). Based on a semi-quantitative assessment of the SIRT1 immunoreactive cells, the SIRT1 protein expression levels correlated inversely with the degree of tumor differentiation (Figure 2). All IRs values are described in Table 1.

### 3.2. Western Blot

Several canine mammary tumor cells were analyzed to determine the endogenous levels of SIRT1 by Western blot analysis (Figure 3). Reactive bands with different signal intensities were observed in the expected molecular weight range (around 120 kDa). Indeed, CMT-U27 cells abundantly expressed SIRT1, whereas MDCK, CMT-U229, CMT-U309, and P114 cells expressed lower levels of this protein, and CF33 and CMT-U131 cells expressed very low levels of SIRT1 (Figure 3). To confirm the cross-reactivity of the antibody, human non-cancer and cancer cells were used as positive controls.

### 3.3. Immunofluorescence

IF showed an absence of SIRT1 immunolabelling in CMT-U309, CF33, and CMT-U131. In CMT-U229 and P114, a weak SIRT1 expression was evident as small granules distributed in the cytoplasm in a perinuclear localization. In CMT-U27, a strong SIRT1 expression was evident as larger cytoplasmic granules (Figure 4). Quantitative analysis was performed on IF results by calculating CTCF using ImageJ software to show the expression of emitted fluorescence. Similar to WB results, the cell lines that had a higher expression of SIRT1 protein were the CMT-U27 (simple ductal carcinoma) and P114 (anaplastic carcinoma) cell lines (Figure 5).

## 4. Discussion

Mammary tumors are among the most common neoplastic lesions in female dogs, and canine mammary cancer is now widely accepted as an animal model for human breast cancer [16,25,26]. Therefore, numerous studies in both human and veterinary oncology have focused on cellular mechanisms of carcinogenesis and regulatory proteins common to either species that may have prognostic and therapeutic value [17]. Sirtuin family proteins, and, specifically, SIRT1, is one of those promising regulatory proteins with highly maintained molecular structure across mammalian species and whose involvement in tumorigenesis and neoplastic progression has been shown [2,10]. SIRT1 expression and function has been studied in human breast cancer [27,28,29,30] and several other tumors in humans [31,32,33,34,35,36,37,38,39,40,41,42], in murine models [43,44], and in cultured tumor cells [45]; however, it has hitherto neither been investigated in canine mammary tumors nor in normal mammary gland tissues. We found SIRT1 regularly expressed in non-neoplastic mammary gland cells in a largely nuclear distribution pattern; this finding is in accordance with the SIRT1 expression pattern in other mammalian tissues [46,47,48,49]. In normal cells, in general, SIRT1 modulates a multitude of pre- and post-transcriptional pathways, interacting mainly with cellular maintenance and survival [50]. However, these essential physiologic functions, in the case of malignancy, may have a dual and still controversial regulatory effect on tumorigenesis: onco-suppressive on one hand, and tumor promoting on the other [10,51]. Morphologically, these regulatory functions of SIRT1 seem linkable to their predominant subcellular localization (cytoplasmic versus nuclear) [52]. Indeed, our results show a characteristic subcellular expression pattern of SIRT1 which seems linked to cellular expression of malignancy. In normal mammary tissue cells, in benign tumor cells, and in low malignancy grade carcinomas (G1 carcinomas), SIRT1 immunoreactivity had strong throughout and was preferentially nuclear; on the other hand, in less differentiated, highly malignant cancer cells (G2–G3 carcinomas), SIRT1 immunoreactivity was generally weaker but shifted to cytoplasmic localization, as shown on tissues in IHC. To further investigate the observed shifting SIRT1 expression from nuclear to cytoplasmic in tissue-derived tumor cells, we employed IF on homologue malignant canine cancer cell lines and noted that, in those malignant cell lines, SIRT1 expression was always cytoplasmic. Nevertheless, there is a caveat, since Bai and Zhang reported possible inconsistencies when determining the subcellular localization of SIRT1 when using different immunoreactivity protocols and/or antibodies [8]. In our hands, however, results were consistent when using IHC on tissue-derived cells and IF on cultured cells. Furthermore, we used, in both protocols, the same antibody and we assured cross-reactivity between human and canine cells by testing human breast cancer cell lines in parallel and by WB. Overall, our findings agree with other studies on different human cancer cells [44]. However, we must point out that the expression of housekeeping (HKP) proteins may vary according to tissue type [53] or other cellular states [54], and that the expression of HKP may change with the density of cultured cells [55]. This relative loss in SIRT1 immunoexpression was also seen in intestinal adenocarcinoma cells by IHC in 22 rhesus macaque primates [56]. Similarly, Song et al. reported nuclear positivity of SIRT1 in normal cells of the human colon and a predominantly cytoplasmic localization shift in colon cancer cells [52]. Although the subcellular localization is not the sole determinant of SIRT1 functions in tumorigenesis, it may account for the dual role of SIRT1 in normal versus neoplastic cells [57,58]. SIRT1 may target its nuclear substrates to exert its tumor suppressor function and target its cytoplasmic substrates to exert its tumor promoter function. We identified his shift and the significantly decreased expression of SIRT1 in the least differentiated canine mammary tumors, which agrees with findings in human cancer [44]; for instance, in human colon cancer, the levels of SIRT1 assessed by IHC were found to be decreasing, which was inversely correlated with tumor progression [37]. Moreover, it has been demonstrated that SIRT1 is involved in DNA damage response, genome stability, and tumor suppression in many other types of cancers in humans, including oral squamous cell carcinoma [59], ovarian cancers [60], and breast cancer [61]. The consistent SIRT1 cellular expression pattern, identified in our malignant tumor samples and characterized by relative loss and cytoplasmic shift, allows us to hypothesize that SIRT1 may be a tumor modulator and may even play an onco-suppressive role in canine mammary tumors. This assumption is supported by a previous study in which in vitro tumor-suppressive effects were found after upregulation of SIRT1 and apoptosis in human oral squamous cell carcinoma by dietary-derived betaine stimulation; selective SIRT1 inhibition, on the other hand, yielded an opposite effect, increasing proliferation of cultured carcinoma cells [62]. Additionally, in one case of a dog with multi-cancer-like syndrome, downregulation of the SIRT1 gene expression was found, enhancing the notion of SIRT1 exhibiting a tumor protecting role [11]. However, on the contrary, some studies are in support of a tumor promoter role of SIRT1: an increased expression of SIRT1 correlated with p53 inactivation, together with apoptosis reduction, was found in human breast cancer cells [63], and, in breast cancer also, SIRT1 upregulation was found by another group, yet this study did not consider any subcellular SIRT1 expression [64]. The controversial role of SIRT1 in tumorigenesis and tumor progression still remains open. However, the results of our pilot study are in support of SIRT1’s tumor suppressor role in canine mammary gland cancer, as we showed, for the first time, reduced immunoexpression and a shift from nuclear to cytoplasmic localization, which was linked to increasing grades of malignancy. However, defining any possible link between cause and effect of SIRT1 in cancerogenesis warrants further investigation.

## 5. Conclusions

The results of our study demonstrated that SIRT1 is regularly expressed in epithelial cells of the canine mammary gland with a largely nuclear distribution pattern. In mammary cancer cells, the expression fades with increasing grades of malignancy and shifts from nuclear to cytoplasmic, as evidenced by immunohistochemistry. This deregulation of SIRT1, observed in canine mammary tumors, might suggest a tumor-modulating role of this regulatory protein, exhibiting in its normal state a possible tumor-protecting function. However, further studies are needed to elucidate SIRT1 pathways in canine mammary gland tumorigenesis, with the final goal to pharmacologically modulate its function, to either enhance its tumor-protecting role or suppress any tumor promoter function.

## Figures and Tables

**Figure 1 animals-13-02609-f001:**
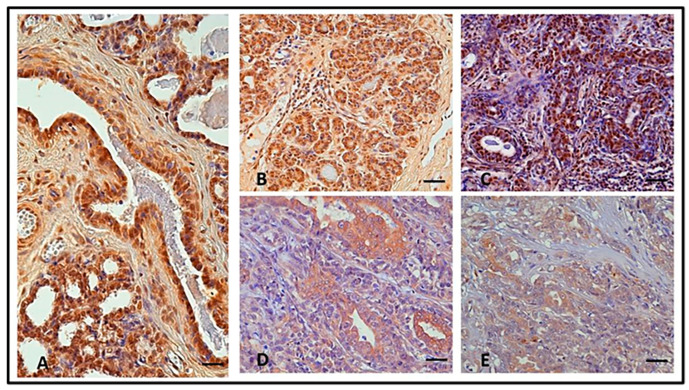
In normal canine mammary gland (NMG), case number 2, strong nuclear SIRT 1 protein expression was found in epithelial ductal cells (**A**). In benign tumor, case number 4, the expression was similar to NMG (**B**). In G1 carcinomas, case number 8, the intensity of immunostaining was strong and localized in the nuclei (**C**). A progressive decrease in IRS values and a loss of nuclear localization was observed in G2, case number 6, (**D**). A further decrease in IRS values and a loss of nuclear localization was observed in G3 case number 7 (**E**). Scale bar: 50 µm.

**Figure 2 animals-13-02609-f002:**
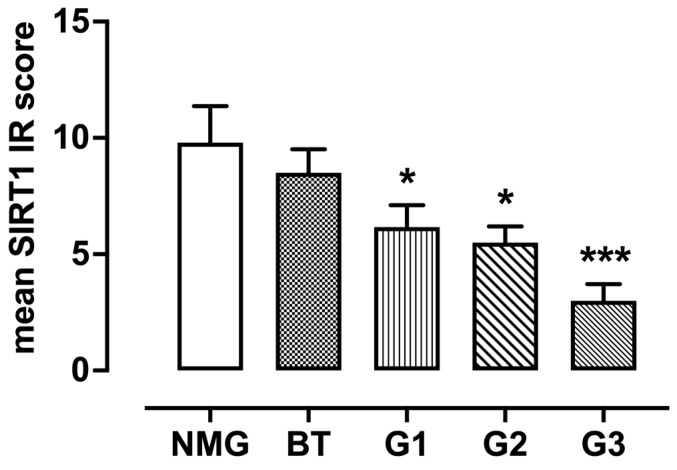
The graph shows the mean ± SEM of the immunoreactive (IR) score for the expression of SIRT1 in normal and tumor samples with different degrees of malignancy. * *p* < 0.05 G1 carcinomas vs NMG; * *p* < 0.05 G2 carcinomas vs. NMG; *** *p* < 0.001 G2 carcinomas vs. NMG; NMG, normal mammary gland; G1, grade 1 carcinoma; G2, grade 2 carcinoma; G3, grade 3 carcinoma.

**Figure 3 animals-13-02609-f003:**
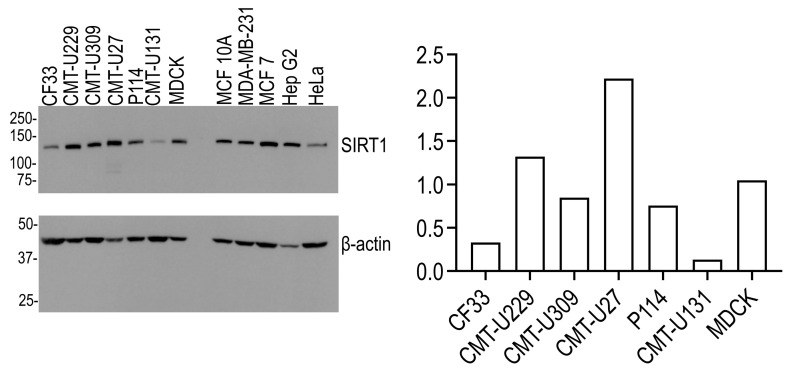
Expression of SIRT1 in CMT cell lines. Western blot analysis of SIRT1 protein expression in different CMT cell lines. The levels of the ß-actin were evaluated to normalize the amount of protein loading. The graphs show the expression of SIRT1 in the CMT cells analyzed.

**Figure 4 animals-13-02609-f004:**
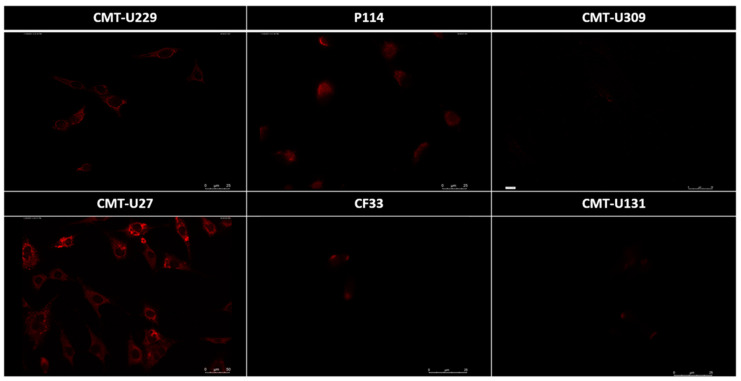
Absence of SIRT1 is evident in CMT-U309, CF33, and CMT-U131 cell lines. Weak SIRT1 expression, characterized by cytoplasmic small granules, is shown in CMT-U229 and P114. Strong SIRT1 expression is evident as larger cytoplasmic granules in CMT-U27.

**Figure 5 animals-13-02609-f005:**
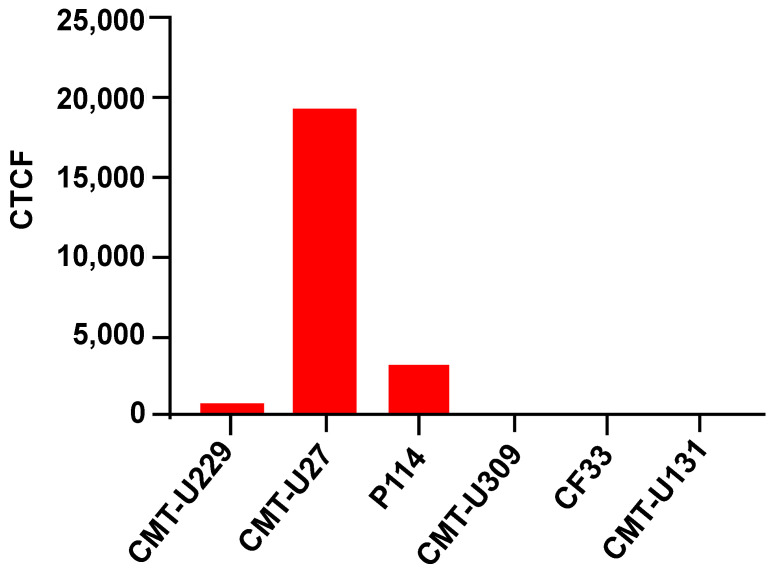
CTCF values, calculated by ImageJ software from Figure 4, quantify the protein expression of SIRT1 in all CMT cell lines used.

**Table 1 animals-13-02609-t001:** IR values for all immunohistochemistry samples.

**Normal Mammary Gland**						
**No. Sample**	**Breed**	**Age (Years)**	**Histologic Classification**	**% Positive Cells**	**Intensity**	**IRS**
1.	Poodle	6	-	78.4	3	12
2.	Mixed breed	9	-	69.5	3	12
3.	Mixed breed	5	-	81.2	3	12
4.	Mixed breed	6	-	27.8	2	4
5.	Jack russel terrier	8	-	32.4	3	9
**Benign Tumor**						
**No. Sample**	**Breed**	**Age (Years)**	**Histologic Classification**	**% Positive Cells**	**Intensity**	**IRS**
1.	Yorkshire terrier	11	Mixed adenoma	58.1	3	9
2.	Poodle	12	Benign mixed tumor	10.8	1	2
3.	Mixed breed	10	Papillary cystic adenoma	39.1	3	9
4.	Maltese	7	Ductal adenoma	38.3	3	9
5.	Mixed breed	8	Tubular adenoma	24.5	3	6
6.	Mixed breed	6	Adenoma Simple	71.4	3	12
7.	Mixed breed	11	Complex Adenoma	66.4	3	12
8.	Mixed breed	6	Adenoma Simple	70.8	2	8
9.	Mixed breed	9	Benign Mixed Tumor	51.3	2	6
10.	Yorkshire Terrier	8	Complex Adenoma	62.6	3	12
**Grade 1**						
**No. Sample**	**Breed**	**Age (Years)**	**Histologic Classification**	**% Positive Cells**	**Intensity**	**IRS**
1.	English setter	8	Mixed type carcinoma	11.3	2	4
2.	Mixed breed	11	Complex type carcinoma	22.5	1	2
3.	Mixed breed	10	Complex type carcinoma	12.3	3	6
4.	Mixed breed	14	Complex type carcinoma	39.8	2	9
5.	Mixed breed	6	Cystic papillary carcinoma	36.1	3	9
6.	Mixed breed	9	Adenocarcinoma tubular	10.1	1	2
7.	Mixed breed	12	Carcinoma arising in benign mixed tumor	24.8	2	4
8.	Mixed breed	10	Simple tubular carcinoma	65.8	3	12
9.	Mixed breed	10	Complex type carcinoma	56.3	3	9
10.	Mixed breed	11	Simple tubular carcinoma	34.3	1	3
11.	Yorkshire Terrier	10	Complex Adenoma	71.6	2	8
12.	Mixed Breed	9	Tubular Carcinoma	27.5	3	6
**Grade 2**						
**No. Sample**	**Breed**	**Age (Years)**	**Histologic Classification**	**% Positive Cells**	**Intensity**	**IRS**
1.	Mixed breed	9	Tubulopapillary carcinoma	15.3	1	2
2.	Poodle	9	Complex type carcinoma	45.1	3	9
3.	Maltese	10	Mixed type carcinoma	18.7	2	4
4.	Mixed breed	11	Solid carcinoma	20	2	4
5.	Mixed breed	10	Tubulopapillary carcinoma	19.2	3	6
6.	Labrador retriever	12	Tubulopapillary carcinoma	31.2	2	6
7.	Mixed breed	9	Complex type carcinoma	37.1	2	6
8.	Shih-tzu	11	Simple tubular carcinoma	9.2	2	2
9.	Mixed Breed	11	Tubular Carcinoma	65.8	2	8
10.	Cocker Spaniel	10	Tubular Carcinoma	56.3	3	9
11.	Epagneul Breton	10	Tubulopapillary Carcinoma	33.8	2	6
12.	Mixed breed	6	Mixed Type Carcinoma	30.4	2	4
**Grade 3**						
**No. sample**	**Breed**	**Age (Years)**	**Histologic Classification**	**% Positive Cells**	**Intensity**	**IRS**
1.	Doberman pinscher	13	Simple tubular carcinoma	40.7	1	3
2.	Maltese	11	Solid Carcinoma	29.2	1	2
3.	Jack Russel Terrier	13	Simple Tubular Carcinoma	27.3	3	6
4.	Greyhound	9	Simple Tubular Carcinoma	6.7	1	1
5.	Siberian Husky	11	Comedocarcinoma	0	0	0
6.	Cocker Spaniel	6	Complex Type Carcinoma	3.4	1	1
7.	Mixed Breed	9	Carcinoma And Malignant Myoepithelioma	13.4	2	4
8.	Mixed Breed	11	Complex Type Carcinoma	43.1	2	6
9.	Shih Tzu	9	Complex Type Carcinoma	0	0	0
10.	Mixed Breed	12	Mixed Type Carcinoma	21.6	3	6
11.	Greyhound	9	Spindle Cell Carcinoma	36.9	2	4

## Data Availability

The data presented in this study is contained within the article.

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
