# Peer review of "Sirtuin 1 Expression in Canine Mammary Tumors: A Pilot Study"

_animals, 2023, doi:10.3390/ani13162609_

Round 1

Reviewer 1 Report

The authors of the publication, entitled "Sirtuin 1 expression in canine mammary tumors: a pilot study," undertook the study of a relatively unknown protein, which may be of considerable, if not major, importance in the future for the elucidation of the mechanisms of mammary tumor formation in bitches. For this reason, it can be said that the work addresses modern issues, performed with advanced research methods in cooperation with many scientific centers mentioned both in the methodology of the work and in the personal acknowledgments to the cooperating scientists. A thorough review of the literature, including two recent papers by one of the authors of this publication, is presented clearly, understandably and the research hypotheses posed are well justified.  However, I do not find unnecessary self-citation here. The results of the study, despite having been performed with several different methods, are understandable and the resulting conclusions are well justified. By virtue of being a reviewer, I have suggestions for changes in the work. I suggest that the affiliation of co-partnering researchers of people from different Universities be moved to the section where thanks are given for their help. In my opinion, the methodology should focus on stating how the planned research was carried out and not on listing the places of work of the co-researchers. However, if this is the custom in this journal, please do not change it. Also, please expand a little on the suggestions for further research in this area and give more limitations on the publication presented for publication.  

Author Response

Response 1: Dear Reviewer, Thank you for your comment and suggestions. We have changed the acknowledgements by deleting them from the Materials and Methods chapter and adding them to the appropriate paragraph at the end of the article. In addition, we have expanded the suggestions for further scientific research with this sentence in conclusion (lines 327-331) “New modulators of SIRT1 have been discovered or synthesized, and clinical trials investigating the therapeutic potential of SIRT1 in cancer treatment seem to show promising results. Therefore, this area of research should be given higher priority, and larger studies should be conducted to decipher the code of dual action of SIRT1 in carcinogenesis and particularly in breast cancer.”, as you suggested. Again, many thanks.

Reviewer 2 Report

Comments on the manuscript animals-2480160-peer-review-v1 entitled “Sirtuin 1 expression in canine mammary tumors: a pilot study” by Sgadari et al.

The manuscript is an interesting investigation on the potential effect of Sirtuin 1 on canine mammary tumorigenesis. New articles on this theme are welcome. Results are interesting, however, there are some changes that must be amended to increase comprehensibility.

Authors should confirm if references in the text are in accordance with ” instructions to the authors” of this journal. 

SIMPLE SUMMARY: 

Lines 13-14: Please replace “healthy” by “non-neoplastic”, and replace “ (5) and neoplastic (45)” to “(n=5) and neoplastic (n=45).”

Lines 19: Please replace “SIRT1expression” to “SIRT1 expression”.

ABSTRACT: 

Lines 26-27: Please amend “canine mammary carcinoma tissue” to “canine mammary tissues” as in addition to carcinomas authors also studied benign and non-neoplastic mammary tissues.

INTRODUCTION 

Lines 41: Please replace “they actively are involved” to “they are actively involved”.

Lines 61-62: Please replace “employing a model we used previously evaluating the carnitine metabolism in cancer (Cacciola NA, 2020) (Cacciola NA., 2021).” To “employing a model previously published (Cacciola NA, 2020) (Cacciola NA., 2021)”, as the mention of previous studies of this investigation group on tumour cell metabolism, are not important for this particular study. 

Lines 72: Please amend “as well ;as to compare SIRT1 expression” to “as well; as to compare SIRT1 expression”.

Lines 74-75: Please amend the following sentence The aim of the present study was to investigate by Western blot analysis (WB). The expression of SIRT1”.

MATERIALS AND METHODS 

Lines 88-90: Please amend “In addition, 5 normal mammary gland (NMG) tissue samples were used as healthy controls, obtained from dogs in which the entire chain of mammary glands had been removed.” to “In addition, 5 non-neoplastic (“normal”) mammary gland (NMG) tissue samples were used as controls…”. 

Authors should also inform if “normal gland” was obtained from animals’ devoid mammary tumours. Mammary gland on the periphery of tumours is not “completely normal” as it is influenced by tumour microenvironment. I assume that if an entire chain was removed, there was at least 1 tumour in the chain…Please clarify.

Lines 97-98: In the sentence “averaged per 10 high power fields (HPF)”, please insert in the manuscript the area of the 40x objective used, as HPF vary with the microscope.

Lines 106-108: Please amend “…method using a commercially available kit as described elsewhere (Cacciola et al., 2020) (streptavidin-biotin-peroxidase method LSAB kit; Dako, Glostrup, Denmark).” to “…method using a commercially available kit (LSAB kit; Dako, Glostrup, Denmark) as described elsewhere (Cacciola et al., 2020).” 

Lines 110-11: Besides negative control, please insert information on the used positive control.

Lines 115-117: Authors should explain why they did not use a previously published evaluation system for this molecule (as Cha et al 2009; Jin et al., 2018; Holloway et al., 2013, …) or similar molecules (Sirt-4 Shi et al., 2016). 

Line 119- Information on Table 1 is irrelevant and table should be removed from the manuscript. Besides, there is an error in the table (>60,1 instead of <60,1)

Lines 127-132: The origin of the cell cultures should be removed from the manuscript and insert in the end of the manuscript in a “Acknowledgments” section.

RESULT SECTION

Lines 201-205: Although there are studies with similar statistical approach (even from your scientific group), are authors sure that is correct to convert a categorical data (both the extent of immunopositivity, and the intensity of immunoreactivity) in a numerical one??

Line 200-203: Please insert information regarding the nuclear positivity on NMG and benign tumours. Although this information is present in Simple Summary (lines 16-17) and in Figure 1 legend (218-220), it is not present in Result section. 

Table 2: Why do authors sometimes use “simple carcinoma”, others “Simple tubular carcinoma”?? Please standardize the criteria. If there are multiple simple tumours histotypes, why highlight the tubular growth pattern?? 

Lines 248 and 256: What authors mean with “larger densities”?? Please clarify. 

DISCUSSION SECTION

Lines 277: Please amend “healthy mammary gland cells” to “non-neoplastic mammary gland cells”.

Lines 321: Please amend “in vitro” to “in vitro”.

REFERENCES

There are multiple errors in references: journal name in full and others in short, missing pages, multiple dots. Please amend.

There are some minor corrections of English language.

Author Response

Dear Editor, Thank you for your suggestions and corrections. I attach the PDF file with the responses to your comments.

Round 2

Reviewer 1 Report

Good work. Thank you

Author Response

Not applicable. 

Reviewer 2 Report

.

.

Author Response

Not applicable.